# A Comparative Analysis of Machine/Deep Learning Models for Parking Space Availability Prediction

**DOI:** 10.3390/s20010322

**Published:** 2020-01-06

**Authors:** Faraz Malik Awan, Yasir Saleem, Roberto Minerva, Noel Crespi

**Affiliations:** CNRS UMR5157, Telecom SudParis, Institut Polytechnique de Paris, 91000 Evry, France; yasir_saleem.shaikh@telecom-sudparis.eu (Y.S.); roberto.minerva@telecom-sudparis.eu (R.M.); noel.crespi@telecom-sudparis.eu (N.C.)

**Keywords:** car parking, decision tree, deep learning, ensemble learning, IoT, K-nearest neighbors (KNN), machine learning, multilayer perceptron, parking sensors, random forest, sensors, smart city, voting classifier

## Abstract

Machine/Deep Learning (ML/DL) techniques have been applied to large data sets in order to extract relevant information and for making predictions. The performance and the outcomes of different ML/DL algorithms may vary depending upon the data sets being used, as well as on the suitability of algorithms to the data and the application domain under consideration. Hence, determining which ML/DL algorithm is most suitable for a specific application domain and its related data sets would be a key advantage. To respond to this need, a comparative analysis of well-known ML/DL techniques, including Multilayer Perceptron, K-Nearest Neighbors, Decision Tree, Random Forest, and Voting Classifier (or the Ensemble Learning Approach) for the prediction of parking space availability has been conducted. This comparison utilized Santander’s parking data set, initiated while working on the H2020 WISE-IoT project. The data set was used in order to evaluate the considered algorithms and to determine the one offering the best prediction. The results of this analysis show that, regardless of the data set size, the less complex algorithms like Decision Tree, Random Forest, and KNN outperform complex algorithms such as Multilayer Perceptron, in terms of higher prediction accuracy, while providing comparable information for the prediction of parking space availability. In addition, in this paper, we are providing Top-K parking space recommendations on the basis of distance between current position of vehicles and free parking spots.

## 1. Introduction

### 1.1. Background

One of the most challenging tasks associated with metropolitan cities like Paris or New York or even smaller ones like Santander, Spain is to find an available parking space. According to an IBM survey [1], about 40% of the road traffic in cities is actually composed of vehicles whose drivers are searching for parking spaces. This problem exacerbates issues such as fuel consumption, pollution emission, road congestion, and wasted time, not to mention contributing to accidents due to the drivers’ main focus on finding their space [2].

Much work has been done on parking space management, e.g., utilizing sensors (for determining available parking spots) [3] and user feedback (i.e., people informing others of parking space availability by means of applications) to identify available parking spaces [4]. Such systems are based on transient data, without the possibility to actually reserve and allocate the parking spots, and so these techniques are only practical in very short timeframes and when the user is in close proximity to the parking areas. Even so, they do not offer any guarantee that a parking spot will be available. However, to predict the availability of free parking spots at a particular time in the future, these systems coupled with Artificial Intelligence (AI)-based approaches can provide solutions. In order to succeed in the task of predicting parking space availability, data generated by the IoT sensors and the IoT devices, combined with ML/DL approaches, can be very useful. Given the variety of ML/DL methods, one technical problem is to identify the most suitable ML/DL model for the problem and the data set, as the performance of each ML/DL model varies from problem to problem and data set to data set. It is important to mention here some of the relevant works that have been done on comparing AI/ML algorithms in several application domains. The use of ML/DL algorithms has been compared for different application fields. For example, Hazar et al. [5] analyze automatic modulation recognition over Rayleigh fading channels. They trained various ML/DL models for this task, including Random Forest, KNN, Artificial Neural Networks (ANN), Support Vector Machines (SVM), Naïve Bayes, Gradient Boosted Regression Tree (GBRT), Hoeffding Tree, and Logistic regression, and found Naïve Bayes to be an optimal algorithm for this problem. While they ranked GBRT and Logistic Regression as the best algorithms in terms of recognition performance, these algorithms required more processing time. Similarly, Naryanan et al. [6] applied Artificial Neural Network (ANN), KNN, and Support Vector Machine (SVM) approaches to a malware classification problem, and found that KNN outperformed SVM and ANN in terms of accuracy.

### 1.2. Contribution

In this paper, we analyze and evaluate various ML/DL models and determine the best predictive model among them for the parking space availability problem using the parking space data set of Santander, Spain. For comparison, we present different ML/DL-based solutions, including KNN, Random Forest, Multilayer Perceptron (MLP), Decision Tree, and a combined model called Voting Classifier (or Ensemble Learning). Although there are many ML/DL techniques available in the literature, we chose these five ML/DL techniques because they are, firstly, well-known and widely used in the community. Secondly, this is a preliminary work which we plan to extend for experimentation and demonstration of the prediction of parking space availability by integrating it into Santander, Spain’s smart parking application for validation and to obtain user feedback. We performed this comparison using the well-known evaluation metrics Precision, Recall, F1-Score, and Accuracy. Our contributions are summarized below with respect to the main objective of predicting the availability of parking spaces:Identification of the best performing, among well-known and generally used ones, AI/ML algorithm for the problem at hand;
-An analysis and evaluation of various ML/DL models (e.g., KNN, Random Forest, MLP, Decision Tree) for the problem of predicting parking space availability;-An analysis/assessment of the Ensemble Learning approach and its comparison with other ML/DL models; and-Recommendation of the most appropriate ML/DL model to predict parking space availability.Recommending top-k parking spots with respect to distance between the current position of vehicle and available parking spots;Application of the algorithms in order to demonstrate how satisfactory prediction of availability of parking spaces can be achieved using real data from Santander;

### 1.3. Impact of Our Parking Prediction Model on Smart Cities

Smart Cities is a widely used term and is an umbrella that accommodates various aspects related to urban research. Mobility and Transportation are considered as the most important branches of the research related to smart cities. Smart transportation and mobility have the potential to make significant contributions in smart cities by utilizing the Internet of Things (IoT) technologies. As described earlier, drivers in search of parking space cause the traffic congestion, affecting many operations and domains of smart cities such as route planning, traffic management, and parking spaces management. Here, the smart parking system makes an effort to reduce the traffic congestion on the roads [7] enriched by our presented parking prediction ML/DL models that makes a significant impact on smart cities. Additionally, since our presented parking prediction models work on the data set of a smart city, Santander, therefore, it can have a direct impact on Santander smart city.

### 1.4. Organization

The organization of this paper is as follows. Section 2 presents the State of the Art. Section 3 provides an overview of the five ML/DL techniques used for our analysis. The performance of these ML/DL techniques is presented in Section 4, and we provide our conclusions and recommendations in Section 5.

## 2. Related Work

Many systems have been proposed to deal with the parking spot recommendation problem. The most common solution to this problem is a recommendation system based on real-time sensors capable of detecting parking space availability [3]. For example, Yang et al. [8] evaluated a real-time Wireless Sensor Network (WSN) linked with a web server that collects the data for determining the available parking spots. These data are then passed on to users by means of a mobile phone application. Similarly, Barone et al. [7] proposed an architecture, named Intelligent Parking Assistant (IPA). The proposed architecture does not provide parking spot availability prediction. In fact, it enables users to reserve a parking spot. In order to reserve a parking spot, the user is supposed to get registered with IPA; only the authorized user can use this architecture. Dong et al. [9] present a simulation-based method, Parking Rank, to deal with the real-time detection of parking spots. Their system collected the public information of parking spots, e.g., price, total available space, rented space, etc. and sorted the parking spots by following the Page Rank algorithm. Since they are based on checking real-time data, these systems do not offer the possibility to predict the availability of a parking space in an area and in a time frame (e.g., between 20 and 30 min from the current time) of interest of the user. Therefore, other solutions have been suggested. A Neural Network based model (MLP) was proposed by Vlahogianni et al. [10] to predict the occupancy rate of parking areas and parking spots. For example, in a specific parking area, there is a 75% probability that a parking space is going to be available in 5 min. Badii et al. [11] performed a comparative analysis of Bayesian Regularized Neural Network, Support Vector Regression, Recurrent Neural Network, and Auto-regressive integrated moving average methods for the prediction of parking spot availability within a specific garage without specifying a particular parking spot. With ML/DL models, there are two different research directions: off-street parking spots and on-street parking spots [11]. Their approach is limited to parking spots inside garages with gates (e.g., off-street parking spots). In addition, they included complex features like weather forecasts in their data set. Zheng et al. [12] performed a comparative analysis of Regression Tree, Neural Network, and Support Vector Regression (SVR) methods for the prediction of parking occupancy rates. Since they were dealing with the occupancy rate, while collecting the data they focused on information such as the number of occupied parking spaces. In terms of predicting the parking occupancy rate, Zheng et al. found that the Regression Tree method outperforms the other two algorithms they evaluated. Camero et al. [13] presented a Recurrent Neural Network (RNN)-based approach to predict the number of free parking spaces. Their main aim was to improve the performance of the RNN. To do so, they introduced a Genetic Algorithm (GA)-based technique and searched for the best configuration for RNN using the GA approach. They utilized the parking data of Birmingham, U.K., which contains the parking occupancy rate for each parking area given the time and date. Yu et al. [14] selected the Auto Regressive Integrated Moving Average (ARIMA) model to predict the number of berths available. ARIMA model is used for making time series forecast. Their experiment was based on a central mall’s underground parking and they collected one month data (October 2010). As this is one month data, we believe it might not give clear insight as the parking occupancy pattern can vary in different months. We believe that different factors like public holidays or other kinds of holidays can affect the performance, so one month data might not be enough to have a clear view. Bibi et al. [15] performed car identification in a parking spot. They collected the video from the camera and divided the parking spots into blocks. Their main contribution is to identify any parking spot it occupied or not using image processing. This processing is being done in real-time and does not provide any future prediction. However, their approach can be used for data collection. Similarly, Tătulea et al. [16] detected the parking spaces and identified if the parking spots are occupied or available using computer vision techniques and the camera as a sensor. In order to do that, they performed different steps, including Frame Pre-processing, Adaptive Background Subtraction, Metrics & Measurements, History Creation, Results Merging for Final Classification, and Parking Space Status. Again, this work is not about the future prediction of parking spots.

In contrast to the above-mentioned works, we deal with the prediction of on-street parking in Santander, a smart city of Spain and our prediction models are based on less complex data features. Moreover, we are targeting individual parking spot’s occupancy status and can make future prediction about such spots with a validity period of 10 to 20 min. Our prediction has a 10 and 20 min validity because, according to our analysis, during peak hours, parking spots near places like city centers or shopping malls usually do not have the same status (free or occupied) for a longer time interval. Their status changes frequently with 10 to 20 min intervals.

## 3. Overview of ML/DL Techniques

Here, we provide an overview of the ML/DP techniques used to evaluate and analyze a data set in order to predict parking space availability. We compared the MLP, KNN, Decision Tree & Random Forest, and Ensemble Learning/Voting Classifier techniques.

### 3.1. Multilayer Perceptron (MLP) Neural Network

MLP is one of the most well-known types of neural networks. It consists of an input layer, one or more hidden layer(s), and an output layer. Each hidden layer consists of multiple hidden units (also called neurons or hidden nodes). The value of any hidden unit *n* in any hidden layer is calculated using Equation (Equation 1) [17]:(1)$$h_{n}=a\left(\sum_{K=1}^{N}i_{K}*W_{K,n}\right),$$
where $h_{n}$ represents the output value of any hidden unit *n* in any hidden layer, and *a* represents the activation function. The activation function is responsible for making the decision related to the activation of a specific hidden unit. *N* in Equation (Equation 1) represents the total number of input nodes (in our case, there are five nodes in the input layer as well as in each of the three hidden layers), and $i_{K}$ represents the value of input node *K* being fed to hidden unit $h_{n}$. This input node can be an input layer node or it can be a node in any previous hidden layer. $W_{K,n}$ represents the weight of unit $h_{n}$. This weight is a measure of the connection strength between an input node and a hidden unit [18].

We used a Rectifier Linear Unit (ReLU) as the activation function for all the layers, so, at each hidden unit, the activation function *a* in Equation (Equation 1), takes the input and returns the output value as follows:(2)$$O_{n}=max\left(0,INP_{n}\right),$$
where $O_{n}$ represents the output value of any hidden unit in any hidden layer, $INP_{n}=\sum_{K=1}^{N}i_{K}*W_{K,n}$ is the input value of any hidden unit in any hidden layer. ReLU function was recommended as an activation function by the grid search approach (Explained in Section 4). Vanishing gradient is one of the major problems faced by DL approaches. Activation functions like Sigmoid and Tanh are not capable of dealing with vanishing gradient problems. However, ReLU does have the ability to deal with vanishing gradient problems [19]. Figure 1 illustrates the concept of a fully connected MLP with three hidden layers and with a number of hidden units equal to the number of features $\left(x_{1},x_{2},\cdots ,x_{n}\right)$ in each sample in the data set. The complete details of these features are provided in Section 4.

### 3.2. K-Nearest Neighbors (KNN)

KNN is known as one of the simplest ML algorithms. It classifies samples on the basis of the distances between them. In any classification data set, there are observations in the form of *X* and *Y* in the training data, where $X_{i}$ is the vector containing the feature values, and $Y_{i}$ is the class label against $X_{i}$. Let us suppose there is an observation $X_{k}$ and we want to predict its class label $Y_{k}$ using KNN. Still using Equation (Equation 3), the KNN algorithm finds the K number of observations in *X* that are close (or similar) to the observation $X_{k}$:(3)$$DIST_{X_{k},X_{i}}=D{\left(X_{k},X_{i}\right)}_{1\leq i\leq n}.$$

Using Equation (Equation 3), the distance between observation $X_{k}$ and all the observations in *X* can be calculated. After calculating these distances, the top-K closest (similar) observations from the training data are selected and then classed as the majority among the top-K closest observations is assigned to unlabeled sample $X_{k}$. There are several distance functions available, including Manhattan, Minkowski, and Euclidean [20]. Euclidean is the most popular; it calculates the distance between observations using Equation (Equation 4):(4)$$D\left(X_{k},X_{i}\right)=\sqrt{\sum_{l=1}^{\#features}\left(X_{l,k}-X_{l,i}\right)},$$
where $X_{l,k}$ represents the $l^{th}$ feature of sample $X_{k}$, and $X_{l,i}$ represents the $l^{th}$ feature of observation $X_{i}$.

### 3.3. Decision Tree and Random Forest

The Decision Tree algorithm constructs a tree by setting different conditions on its branches. An exemplary architecture of a Decision Tree is shown in Figure 2. It consists of (i) a root node (i.e., the starting point), (ii) internal nodes (where splitting takes place), and (iii) leaves (Terminal or Final Nodes that contain the homogeneous classes). Again considering the same scenario, with *X* as the training data set, *N* as the total number of observations and their corresponding class labels (*C*) in *X*, the entropy can be calculated using Equation (Equation 5) [21]:(5)$$E\left(X\right)=-\sum_{j-1}^{K}\frac{freq(C_{j},X)}{N}\log_{2}\frac{freq(C_{j},X)}{N},$$
where $\frac{freq(C_{j},X)}{N}$ represents class $C_{j}$’s occurrence probability in *X*, and *N* represents the total number of samples in the training set. The information gain is then used to perform node split using equations given in [21].

The Random Forest algorithm is similar to the Decision Tree algorithm. In fact, it consists of multiple independent Decision Trees. Each tree in a Random Forest sets conditional features differently. When a sample arrives at a root node, it is forwarded to all the sub-trees. Each sub-tree predicts the class label for that particular sample. At the end, the class in the majority is assigned to that sample.

### 3.4. Ensemble Learning Approach (Voting Classifier)

Figure 3 illustrates the concept of Voting Classifier, also known as the Ensemble Learning Approach that combines multiple ML/DL models. In this paper, we combined MLP, KNN, Decision Tree, and Random Forest algorithms to solve the problem of predicting the availability of parking spaces. The Ensemble Learning approach takes the training data and trains each model. After the training process, the Ensemble Learning approach feeds the testing data to the models and then each model predicts a class label for each sample in the testing data. In the next stage, a voting process is performed for each sample prediction. Generally, two kinds of voting are available: hard voting and soft voting. In *hard voting*, the Ensemble Learning approach assigns a class label, voted by majority, to the sample. For example, among five models, three models identify that the same sample $X_{k}$ belongs to Class $C_{1}$ while the other two models identify that this sample belongs to Class $C_{2}$. Given that Class $C_{1}$ has been voted for by the majority, Class $C_{1}$ would be assigned to that particular sample.

*Soft voting*, on the other hand, averages the probability of all the expected outputs, i.e., the class labels, and then the class with the highest probability is assigned to the sample.

## 4. Results and Evaluation

During this work, the algorithms described in the previous section have been used, fine-tuned, analyzed and compared with respect to the specific goal of the recommendation system: i.e., to suggest drivers the most probable and closest location to their final destination for a free parking space by looking ahead in a specific time frame (e.g., 20 min times frame). Data were collected by sensors deployed in a real environment, i.e., the smart city Santander. In this section, we evaluate the performance of five ML/DL models for the prediction of parking space availability and provide a comparative analysis of the preliminary results, which we plan to extend by integrating them into a smart parking application for Santander, Spain for future experimentation.

### 4.1. Parking Space Data Set

The data set for the prediction was obtained by collecting the measurements of sensors deployed in Santander, a smart city in Spain. Almost 400 on-street parking sensors are deployed in the main parking areas of the city center. These parking sensors [22] capture the status (i.e., occupied or free) of the parking spots. Collected over a 9-month period, this data set was constructed as part of the WISE-IoT [23], an H2020 EU-KR project. In WISE-IoT, the parking sensor data was stored in an Next Generation Service Interface (NGSI) context broker [24]. We accessed real-time Santander data; in the WISE-IoT project, in order to make data more consistent, we created a script that retrieves and stores the on-street parking sensor data every minute. The objective is twofold: to predict the parking spot availability within a time interval (validity) of 10 to 20 min, and to evaluate the prediction accuracy. The collected data set has around 25 million records. We conducted our initial experiment using data set having around 3 million records. Later on, in order to check the impact of larger data set on the algorithms, the data set was extended to 25 million records. As scaling up the data set size did not affect the standing (ranking) of ML/DL algorithms, we present the results for 25 million records in the Performance Evaluation section. The collected data set has the following organization:**Parking ID:** Refers to the unique ID associated with each parking space.**Timestamp:** The Timestamp of the parking space data collection.**Start Time/End Time:** Start Time and End Time refer to the time interval during which a parking space’s status remained the same, i.e., available or occupied.**Duration:** Refers to the total duration in seconds during which a specific parking space remained available or remained occupied.**Status:** This feature represents the status of a parking space, e.g., available or occupied.

The above-mentioned features were further organized to be input features for our ML/DL model, as given in Table 1.

Start hour, start minute and end hour, end minute in Table 1 present the 10 or 20 min interval status for any particular parking spot. We collected our data set after every minute; therefore, in order to get the 10 and 20-min status and to provide predictions with 10 and 20 min validity, we used 10 and 20-min windows with 60% and 80% thresholds. For example, if specific parking spot had a 60% availability rate, then, for that 10 or 20-min window, the status of that particular parking spot would be considered available (Free). Similarly, for an 80% threshold, a parking spot would need to have an 80% availability rate in a 10- or 20-min window to be classified “Free”.

### 4.2. Hyper-Parameters of ML/DL Techniques

Table 2 presents the hyper-parameters of the five ML/DL models that we tuned for our comparative analysis. We used GridSearch [25] in order to get the best hyper-parameters’ values for each Machine/Deep Learning model. For MLP, we tuned four hyper-parameters. “Activation” is responsible for determining how active a specific neuron (hidden unit) is. We adopted the widely-used ReLU activation function. As shown in Equation (Equation 2), it returns either 0 or the input itself, and then selects the maximum value between 0 and the input value. This means that, if the input value of a neuron is negative, it will return 0 to keep the output of a neuron within range [0, input value]. “hidden_layer_sizes” defines the number of hidden layers and the number of neurons in each hidden layer.

In our case, its value is (5,5,5), which shows that three hidden layers with five neurons in each layer are being used in the network. We used some rules of thumb [26] to determine the range of the hidden layer sizes and the neuron sizes. The hyper-parameters “learning_rate” and “learning_rate_init” are responsible for the optimization and minimization of the loss function. We used the “adaptive” learning rate. When the learning rate is set to “adaptive”, it keeps the learning rate constantly equal to the initial learning rate as long as there is a decrease in the training loss in each epoch. Every time two consecutive epochs fail to show a decrease in loss function by at least “tol” (tolerance, a float variable for optimization, we used its value = 0.0001), the learning rate is divided by 5. Similarly, for KNN, we tuned four hyper-parameters (i) n_neighbors; (ii) distance metric (Euclidean); (iii) n_jobs (Parallel jobs in search of the nearest neighbors); and (iv) weights (when this is set to uniform all neighboring points are weighted equally). Initially, we did experiments with different numbers of neighbors (1, 5, 7, 11, 25, 50 and 100). “n_neighbors = 11” proved the best option. (Later on, GridSearch also suggested 11 as an optimized parameter). We tuned three hyper-parameters for a Decision Tree. “max_depth” defines the maximum depth of the tree. When it is set to “None”, nodes keep expanding until all the leaves end up having only one class in them, or until all the leaves have samples less than min_samples_split in them. However, having a Decision Tree that is too deep could lead to the problem of overfitting. “min_samples_split” represents the minimum number of samples required for a node to go for a further split. Similarly, “min_samples_leaf” defines how many samples a leaf node can contain. “criterion = entropy” works on information gain, which is the information related to the decrease in entropy after a split. “n_estimators” defines the number of trees in the forest. Its default value is 10. As we have a huge data set (∼25 million records), we keep the number of estimators close to the usually-recommended range for a huge data set (i.e., 128 to 200). For an Ensemble Learning approach, the hyper-parameter “estimators” defines the ML/DL models to be used for prediction, while the hyper-parameter “weights” defines the priority given to each estimator. We assigned equal weights to all the estimators except Decision Tree. We gave Decision Tree a higher priority, as it performed relatively better than the rest of the ML/DL models when it was used alone for parking space prediction. The hyper-parameter “voting” is described in Section 3.

### 4.3. Evaluation Metrics

The performance metrics we used for the evaluation and comparison of ML/DL models are given below. Moreover, to check the overfitting and stability of these models, we performed K-fold cross-validation. Each evaluation metric and K-fold cross-validation are explained below:*Precision* can be defined as the fraction of all the samples labelled as positive and that are actually positive [27]. It can be mathematically presented as follows:
(6)$$Precision=\frac{TruePositive}{TruePositive+FalsePositive}.$$*Recall*, in contrast, is defined as the fraction of all the positive samples; they are also labeled as positive [27]. Mathematical presentation of recall is given below:
(7)$$Recall=\frac{TruePositive}{TruePositive+FalseNegative}.$$The *F1-Score* is defined as the harmonic mean of recall and precision [27], defined mathematically as:
(8)$$F1\text{-}Score=\frac{2*(Recall*Precision)}{Recall+Precision}.$$*Accuracy* is the measure of the correctly predicted samples among all the samples, expressed in an equation as:
(9)$$Accuracy=\frac{\#CorrectPredictions}{\#TotalSamples}.$$*K-fold cross-validation* is a method for checking the overfitting and evaluating how consistent a specific model is. In K-fold validation, a data set is divided into *K* equal sets. Among those *K* sets, each set is used once as testing data and the remaining sets are used as training data. In this paper, we used 5-fold cross-validation.

### 4.4. Performance Evaluation

This section provides an evaluation of the performance of the MLP, KNN, decision tree, random forest, and Ensemble Learning algorithms in terms of scores related to each cross-validation. A comparative analysis for 10-min and 20-min prediction was done, considering 60% and 80% thresholds for both predictions.

#### 4.4.1. 10-Min Prediction Validity (60% Threshold)

Table 3 presents the average cross-validation score of MLP, KNN, Random Forest, Decision Tree, and Ensemble Learning models given 10-min predictions with a 60% threshold. It can be seen that the computationally complex model, MLP, showed the lowest performance with an average of 64.63% precision, 52.09% recall, 57.68% F1-Score, and 70.48% accuracy. In contrast, one of the simplest ML models, KNN, outperformed MLP with the results of 73.04% precision, 67.46% recall, 70.14% F1-Score, and 76.71% accuracy. Random Forest performed even better, with 86.90%, 80.11%, 83.37%, and 86.50% for average precision, recall, F1-Score, and accuracy, respectively. Decision Tree’s and Ensemble Learning’s performances were quite close to each other. Decision Tree showed 91.12% average precision while Ensemble learning had 92.79% average precision. The average recall scores for Decision Tree and Ensemble Learning were 90.28% and 89.24%, respectively. The average F1-Score for Decision Tree was 90.69% while Ensemble Learning showed 90.98%. The average accuracy for Decision Tree was 92.25%, while Ensemble Learning, despite combining all the models, could achieve 92.54% accuracy, an improvement of only 0.29%.

#### 4.4.2. 10-Min Prediction Validity (80% Threshold)

Table 4 presents the average cross-validation scores of the ML/DL models given a 10-min prediction validity with an 80% threshold. Following the 60% threshold trend, MLP performed the worst among all the models being compared. MLP showed 70.48% average accuracy with 64.63% average precision, 52.09% average recall, and 57.68% average F1-Score. KNN had a 76.71% average accuracy, 73.04% average precision, 67.46% average recall, and a 70.71% average F1-Score. Random Forest’s average accuracy was 86.50% while its average precision, recall, and F1-Score were 86.90%, 80.11%, and 83.37%, respectively. Again, Decision Tree and Ensemble Learning showed quite similar performances, both at the top end. The average accuracy for Decision Tree and Ensemble Learning was 92.39% and 92.60%, respectively. The average precision shown by Decision Tree was 91.11%, while it was 93.01% for Ensemble Learning. Recall and F1-Score for Decision Tree were 90.32% and 90.71%, respectively. For Ensemble learning, average recall was 88.87% and average F1-Score was 90.89%.

#### 4.4.3. 20-Min Prediction Validity (60% Threshold)

In this section, we present the comparative analysis given a 20-min predication validity with a 60% threshold. Table 5 presents the average cross-validation score for each model. MLP, the lowest scorer overall, showed 64.97% and 52.16% for precision and recall, respectively. With the F1-Score being dependent on precision and recall, MLP’s remained low at 57.83%. MLP’s average accuracy was 70.83%. The performance of KNN remained better than that of MLP. It showed 74.15% in average precision, 68.76% for average recall, 71.35% as its average F1-Score, and 77.71% for average accuracy. Random Forest again performed better than these first two, with 82.44% in average precision, 73.78% for average recall, 77.87% as its average F1-Score, and 82.49% for average accuracy. Decision Tree and Ensemble Learning, following their earlier trend, gave very similar performances. Average accuracy for Decision Tree and Ensemble Learning was 87.66% and 88.73%, respectively. The average precision and average recall shown by Decision Tree were 85.64% and 84.37%, respectively, while these were 88.65% and 83.56% for Ensemble Learning. The F1-Scores for Decision Tree and Ensemble Learning were 85% and 86.03%, respectively.

#### 4.4.4. 20-Min Prediction Validity (80% Threshold)

Here, we present the evaluation results of all the ML/DL models given a 20-min prediction validity and an 80% threshold.

Table 6 shows that, as with the previous experiments, the threshold value did not affect the standing of ML/DL models for this configuration (prediction validity of 20 min and an 80% threshold). Decision Tree and Ensemble Learning remained the top two performers in terms of all evaluation metrics. Ensemble Learning showed 89.02%, 82.52%, 85.64%, and 88.70% for average precision, recall, F1-Score, and accuracy, respectively, and Decision Tree had 85.42% average precision, 84.13% average recall, 84.77% average F1-Score, and 87.82% average accuracy. Random Forest, as the next best, showed 82.86%, 73.56%, 77.93%, and 83.15% for average precision, recall, F1-Score, and accuracy, respectively. KNN, again outperforming lowest-ranked MLP, showed 74.36% for average precision and 68.35% for average recall, with 71.24% and 78.38% for its F1-Score and accuracy, respectively. MLP, being the worst performer, had results of 65.33%, 51.83%, 57.80%, and 72.07% for average precision, recall, F1-Score, and accuracy, respectively.

For a better view, Figure 4, Figure 5, Figure 6 and Figure 7 present the graphical comparison of all the models. By analyzing all the experimental results, it is clear that, in terms of the evaluation metrics, Decision Tree and Ensemble Learning performed better than the other models. However, given the complexity of the Ensemble Learning approach (a combination of all the models), it did not show a significant improvement when compared to the Decision Tree model. When both computational complexity and performance are considered, Decision Tree was the optimized model throughout all of these experiments.

#### 4.4.5. Training Data Evaluation

The size of a training data set can significantly influence the performance of an ML/DL Model. Therefore, in order to further evaluate all five ML/DL models, we performed another comparison, designed to observe how the size of the parking space training data set affects the performance of these models. We chose a subset of the total data set containing 1,252,936 records. We partitioned this data set into five equal folds and set one of the folds as the testing data. Hence, each fold contains 250,587 records. To ensure better observations, we began training our models with a very small number of records: 1000 records. We then added the next 40,000 records and trained the models with 50,000 records. For the third iteration, we trained the models with the 250,587 records of one fold. Then, for the rest of the iterations, we added 250,587 records into the training data set to keep flow consistent. Not following the trend of the other iterations, for the first two iterations, training data set size was randomly chosen as very low (1000, 50,000) to observe how the models behave with very low training data size.

We evaluated the performance of each model in terms of accuracy, gradually increasing the training data size at each level (Figure 8). Figure 9 shows that, after 50,000 records, KNN and Random Forest have a constant, very low increase in accuracy, leading to very moderate improvement. In contrast, Decision Tree and Ensemble Learning showed a bit lower accuracy (around 64% and 68%, respectively) when 1000 records were used as training data. However, both of these models showed continuous improvement as more data were added to the training set. MLP, in contrast, showed a very low accuracy (around 28%) when 1000 records were used, and then only a negligible improvement (almost no improvement) from its accuracy at 50,000 records.

We conducted this experiment for the scenarios mentioned in the *Performance Evaluation* section and found a similar behavior throughout. This experiment, based on a subset of the data set, reveals the behavior exhibited by these models when used for the scenarios in the *Performance Evaluation* section.

#### 4.4.6. Distance Based Recommendation

Individual parking spot prediction enables us to recommend the parking spot to the user with respect to distance. As shown in Figure 10, all the parking spots, predicted as “available”, can be sorted with respect to distance given position (coordinates) of vehicle and parking spots. Users cannot be given indications of parking spaces too far from each other. Thus, the calculation and the clustering of results should be organized by identifying some limited areas close to the final destination that have the highest probability to have free parking spaces. For the time being, no reservation capabilities nor differentiation between prices of the parking slot have been considered; however, these are functions that could be easily added to a recommendation system.

In order to calculate the distance between vehicle and free parking spots and provide recommendation on the basis of distance, we use GPS coordinates of parking sensors & vehicle, and the well-known Haversine formula [28]:(10)$$a=\sin^{2}\left(\delta \theta /2\right)+\cos\theta_{1}-\cos\theta_{2}\sin^{2}\left(\delta \lambda /2\right),$$
(11)$$c=2a\tan2(\sqrt{a},\sqrt{1-a}).$$
(12)Distance=R.c

In Equations (Equation 10)–(Equation 12), θ is latitude, λ is longitude, and *R* is earth’s mean radius (i.e., 6371 km). After calculating the distance between vehicle and free parking spots, sensors are sorted in ascending order (from nearest to farthest). A functionality, built on such calculation, can be used to recommend to users the closest parking slot with the maximum probability of finding it free.

## 5. Conclusions

The analysis took into consideration some of the well-known and most used algorithms, newer or emerging ones could be considered and analyzed in further studies. The novelty of the study is related to the compared analysis of them on the basis of data sets of different sizes but containing data reflecting the real environment. Our goal was to find the optimized Machine/Deep Learning model for the prediction of parking space availability by performing comparative analysis of five different well-known Machine/Deep Learning Models: Multilayer Perceptron (MLP), K-Nearest Neighbors (KNN), Decision Tree, Random Forest, and the Voting Classifier/Ensemble Learning (EL) approach. This paper presents the numerical results based on K-fold cross-validation. Precision, Recall, F1-score, and Accuracy were used as evaluation metrics. We conducted experiments to predict the availability of parking spots with 10- and 20-min prediction validity, setting 60% and 80% as availability thresholds. These features can be tuned according to the needs of users and the specific experience of the service to provide to users. These values were considered meaningful and useful in an environment such as Santander. One of the main contributions of this paper is that it seeks to evaluate if a better result can be produced for the parking space availability prediction problem by using less complex algorithms. From the results of our comparative analysis, we found that *Decision Tree* is the optimal solution for the parking space availability prediction problem, and that Ensemble Learning was a close second best model. With this comparison, we observed that one of the simplest algorithms (KNN) consistently outperformed one of the computationally complex algorithms (Multilayer Perceptron). We also conducted experiments to observe the affect of training data size on all five of the ML/DL algorithms compared in this paper. We plan to extend our work to (i) demonstrate the efficiency of the Decision Tree model by integrating it into the smart parking application of Santander, Spain and obtain user feedback, and (ii) use the Santander, Spain road traffic data set and offer recommendations for parking spot management based on traffic data. A recommendation system can integrate the prediction functionality by adopting the algorithm that is better aligned and predicts results with the needed precision. On this basis, additional features and functions can be added in order to improve the customer experience. Some features can be devoted to improve and simplify the search for an available parking space; however, in conjunction with the government of the city or considering some pollution related considerations, some novel policies for directing people to the “right” destination could be considered, implemented, and verified in the field.

## Figures and Tables

**Figure 1 sensors-20-00322-f001:**
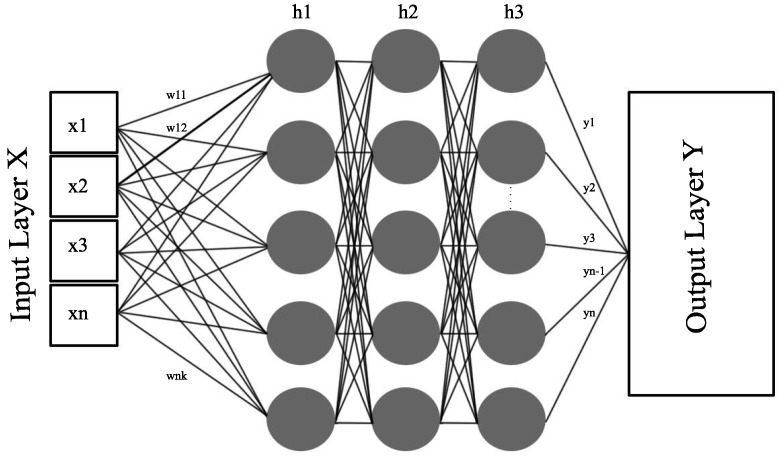
MLP architecture.

**Figure 2 sensors-20-00322-f002:**
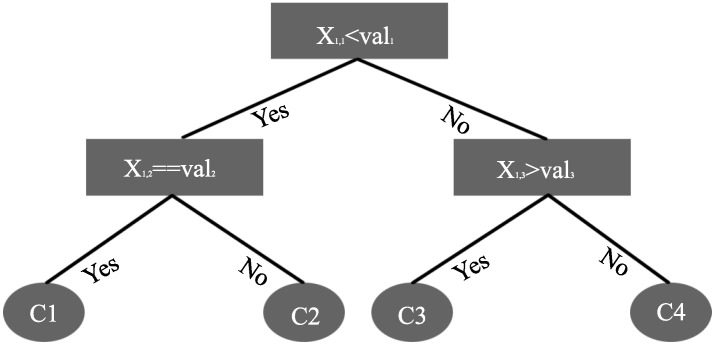
Decision tree architecture.

**Figure 3 sensors-20-00322-f003:**
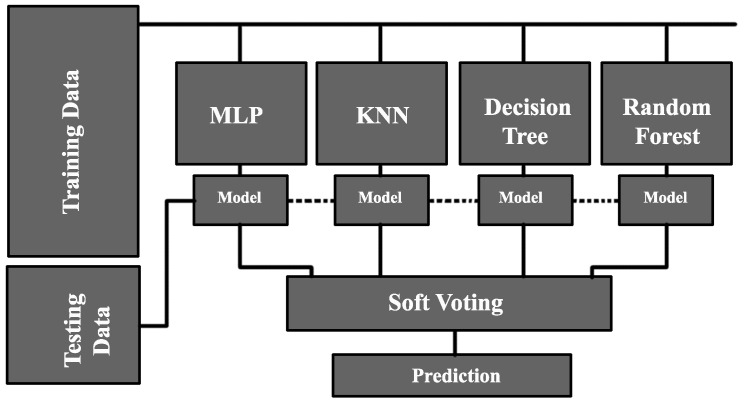
Ensemble Learning or Voting Classifier Architecture.

**Figure 4 sensors-20-00322-f004:**
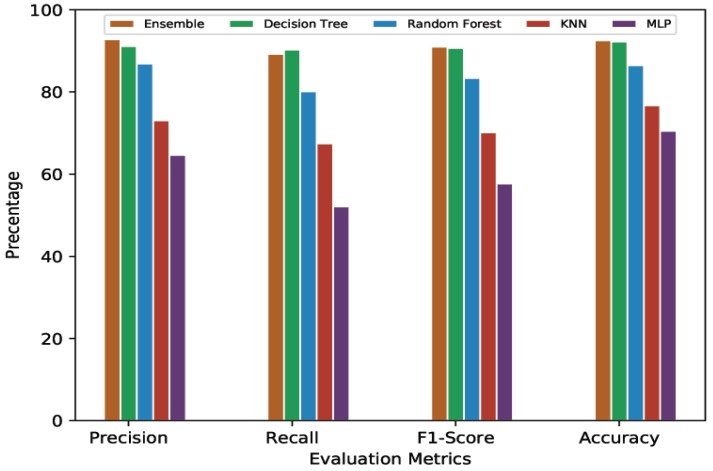
Graphical representation of comparative analysis of ML/DL approaches (prediction validity = 10 min, threshold = 60%).

**Figure 5 sensors-20-00322-f005:**
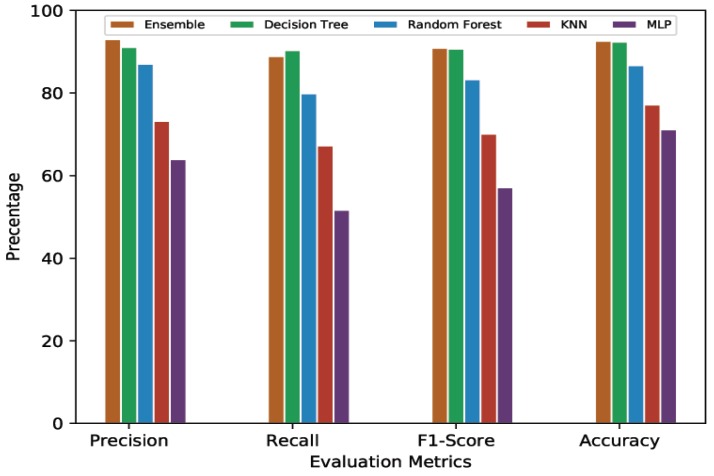
Graphical representation of comparative analysis of ML/DL approaches (prediction validity = 10 min, threshold = 80%).

**Figure 6 sensors-20-00322-f006:**
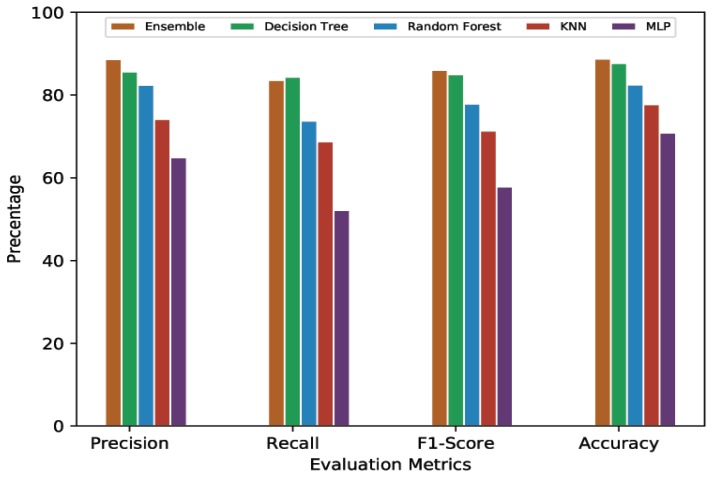
Graphical representation of comparative analysis of ML/DL approaches (prediction validity = 20 min, threshold = 60%).

**Figure 7 sensors-20-00322-f007:**
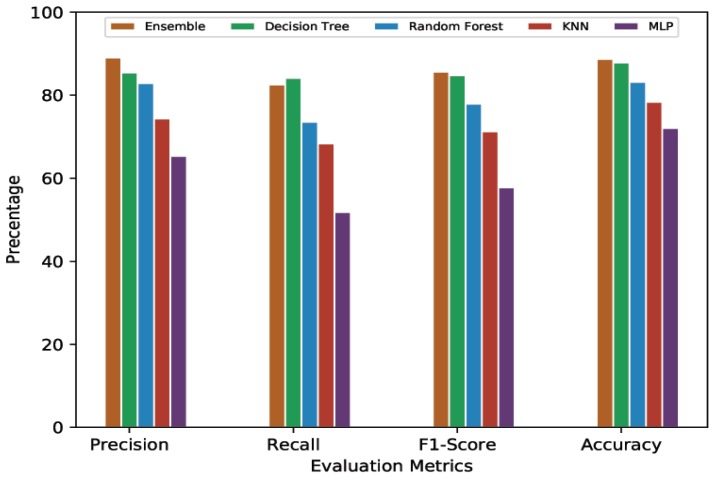
Graphical representation of comparative analysis of ML/DL approaches (prediction validity = 20 min, threshold = 80%).

**Figure 8 sensors-20-00322-f008:**
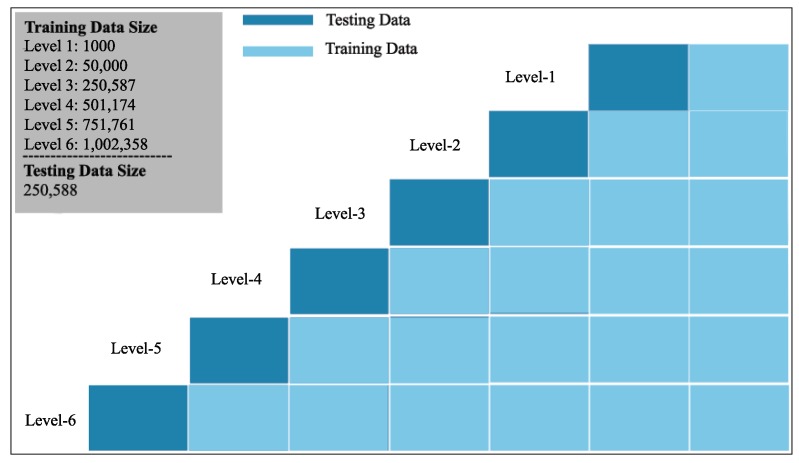
Training data size evaluation method.

**Figure 9 sensors-20-00322-f009:**
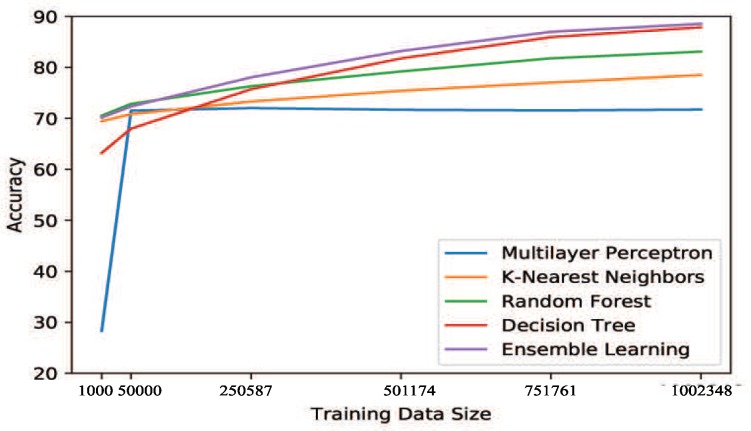
Performance evaluation of training data size.

**Figure 10 sensors-20-00322-f010:**
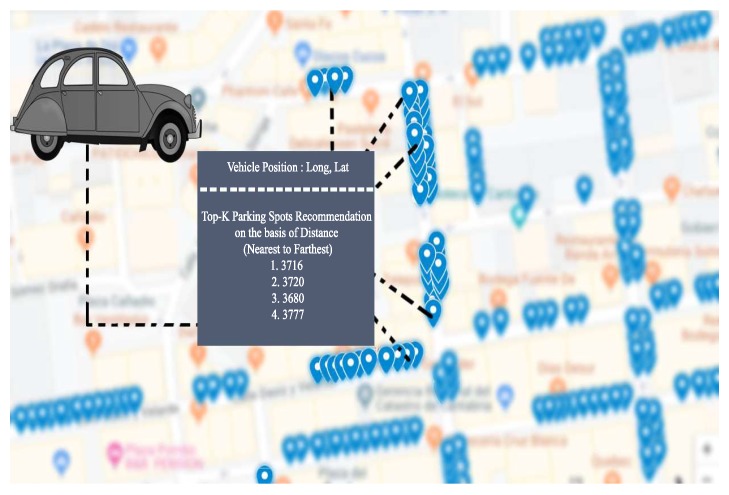
Recommending top-K parking spots on the basis of distance.

**Table 1 sensors-20-00322-t001:** Extracted features.

Features	Value/Range
Parking Spot ID	Unique ID of Sensor
Day	1–7 (Day of the week)
Start Hour	0–23
Start Minute	0–59
End Hour	0–23
End Minute	0–59
Status	0–1 (Occupied or Free)

**Table 2 sensors-20-00322-t002:** Hyper-parameters of ML/DL techniques.

MLP		KNN		Decision Tree		Random Forest		Voting Classifier	
**Parameter**	**Value**	**Parameter**	**Value**	**Parameter**	**Value**	**Parameter**	**Value**	**Parameter**	**Value**
activation	ReLU	n_neighbors	11	max_depth	100	max_depth	100	estimators	MLP, KNN, Random Forest, Decision Tree
early_stopping	True	metric	euclidean	criterion	entropy	criterion	entropy	voting	soft
hidden_layer_sizes	(5,5,5)	n-jobs	None	min_samples_leaf	5	min_samples_leaf	1	weights	1,1,1,2
learning_rate	Adaptive	weights	uniform			n_estimators	200		
learning_rate_init	0.001								
solver	sgd								
tol	0.0001								

ML = Machine Learning, DL = Deep Learning, MLP = Multilayer Perceptron, KNN = K-Nearest Neighbors.

**Table 3 sensors-20-00322-t003:** Average cross validation score of each model (10-min prediction validity with a 60% threshold).

Metrics	MLP	KNN	RF	DT	EL
**Precision**	64.63	73.04	86.90	91.12	92.79
**Recall**	52.09	67.46	80.11	90.28	89.24
**F1-Score**	57.68	70.14	83.37	90.69	90.98
**Accuracy**	70.48	76.71	86.50	92.25	92.54

MLP = Multilayer Perceptron, KNN = K-Nearest Neighbors, RF = Random Forest, DT = Decision Tree, EL = Ensemble Learning.

**Table 4 sensors-20-00322-t004:** Average cross validation score of each model (10-min prediction validity with 80% threshold).

Metrics	MLP	KNN	RF	DT	EL
**Precision**	63.92	73.19	87.01	91.11	93.01
**Recall**	51.64	67.23	79.86	90.32	88.87
**F1-Score**	57.13	70.08	83.28	90.71	90.89
**Accuracy**	71.14	77.18	86.70	92.39	92.60

MLP = Multilayer Perceptron, KNN = K-Nearest Neighbors, RF = Random Forest, DT = Decision Tree, EL = Ensemble Learning.

**Table 5 sensors-20-00322-t005:** Average cross validation score of each model (20-min prediction validity with a 60% threshold).

Metrics	MLP	KNN	RF	DT	EL
**Precision**	64.87	74.15	82.44	85.64	88.65
**Recall**	52.16	68.76	73.78	84.37	83.56
**F1-Score**	57.83	71.35	77.87	85.00	86.03
**Accuracy**	70.83	77.71	82.49	87.66	88.73

MLP = Multilayer Perceptron, KNN = K-Nearest Neighbors, RF = Random Forest, DT = Decision Tree, EL = Ensemble Learning.

**Table 6 sensors-20-00322-t006:** Average cross validation score of each model (20-min prediction validity with an 80% threshold).

Metrics	MLP	KNN	RF	DT	EL
**Precision**	65.33	74.36	82.86	85.42	89.02
**Recall**	51.83	68.36	73.56	84.13	82.52
**F1-Score**	57.80	71.24	77.93	84.77	85.64
**Accuracy**	72.07	78.38	83.15	87.82	88.70

MLP = Multilayer Perceptron, KNN = K-Nearest Neighbors, RF = Random Forest, DT = Decision Tree, EL = Ensemble Learning.

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
