# Peer review of "A Comparative Analysis of Machine/Deep Learning Models for Parking Space Availability Prediction"

_sensors, 2020, doi:10.3390/s20010322_

Round 1

Reviewer 1 Report

The paper provides a comparison between different ML/DL approaches for parking prediction. The simulation demonstrates the performance differences between those approaches.

The text should be checked through to avoid any typos and errors, such as: page 4, "are provided in Section IV", actually there is no "Section IV" in the paper. Maybe section 4? etc. 

Author Response

Comment: The paper provides a comparison between different ML/DL approaches for parking prediction. The simulation demonstrates the performance differences between those approaches. The text should be checked through to avoid any typos and errors, such as: page 4, "are provided in Section IV", actually there is no "Section IV" in the paper. Maybe section 4? etc.

Response:

We are very thankful to the reviewer for the comments and highlighting this mistake which was overlooked by us. They helped us a lot to improve the quality of our paper. We have now carefully proofread the paper and corrected all the typos and errors. We also have corrected the mistake of ‘Section IV’ by changing it to ‘Section 4’, as well as all the similar mistakes. Now all the typos and errors in this manuscript have been corrected. We have provided an updated version that contains blue color text which shows updated/newly added text in the revised manuscript and red color strikethrough text, which shows removed text in the original submission. 

We provide a summary of changes below:

Section ‘1. Introduction’

The organization of this paper is as follows. Section II2 presents the State of the Art. Section III3 provides an overview of the five ML/DL techniques used for our analysis. The performance of these ML/DL techniques is presented in section IV4, and we provide our conclusion and recommendations in section V5.

Section ‘3.1. Multilayer Perceptron (MLP) Neural Network’

ReLU function was recommended as an activation function by the grid search approach (Explained in Section Section IV 4).).

The complete details of these features are provided in Section IV4.

Section ‘4.2. Hyper-Parameters of ML/DL Techniques’

The hyper-parameter "voting" is described in section III3.

Reviewer 2 Report

The authors address an interesting problem and perform a sound comparison among different approaches for parking space availability prediction. however, the effect of such a model on the environment has been overlooked. I would suggest the authors consider the following work and with respect to it dedicate a small section of their introduction to the impact of smart parking management in smart cities.

R. E. Barone, T. Giuffrè, SINISCALCHI S. M., Morgana M. A., and Tesoriere G. (2014). Architecture for parking management in smart cities. IET INTELLIGENT TRANSPORT SYSTEMS, vol. 8; p. 445-452.

Author Response

Comment: The authors address an interesting problem and perform a sound comparison among different approaches for parking space availability prediction. however, the effect of such a model on the environment has been overlooked. I would suggest the authors consider the following work and with respect to it dedicate a small section of their introduction to the impact of smart parking management in smart cities.

Response: 

We are very thankful to the reviewer for the careful reading and highlighting our contributions and main points in this paper, as well as appreciating our paper. We are also thankful for this suggestion of adding a new section of the effect of our model on the environment and providing a reference for it. We have provided an updated version that contains blue color text which shows updated/newly added text in the revised manuscript and red color strikethrough text, which shows removed text in the original submission. 

We have now added a new section in our Introduction section as follows:

New Added Section

1.3. Impact of our Parking Prediction Model on Smart Cities

Smart Cities is a widely used term and is an umbrella that accommodates various aspects related to urban research. Mobility and Transportation are considered as the most important branches of the research related to smart cities. Smart transportation and mobility have the potential to make significant contributions in smart cities by utilizing the Internet of Things (IoT) technologies. As described earlier, drivers in search of parking space cause the traffic congestion, affecting many operations and domains of smart cities such as route planning, traffic management and parking spaces management. Here, smart parking system makes an effort on reducing the traffic congestion on the roads [7] enriched by our presented parking prediction ML/DL models that makes a significant impact on smart cities. Additionally, since our presented parking prediction models work on real data set of a smart city, Santander, therefore, it can have a direct impact on Santander smart city.